# *In vivo* cisplatin-resistant neuroblastoma metastatic model reveals tumour necrosis factor receptor superfamily member 4 (TNFRSF4) as an independent prognostic factor of survival in neuroblastoma

Catherine Murphy[1,2,3], Laura Devis-Jauregui[4], Ronja Struck[1,2,3], Ariadna Boloix[5], Ciara Gallagher[1,2,3], Cian Gavin[1,3], Federica Cottone[1,2,3], Aroa Soriano Fernandez[5], Stephen Madden[6], Josep Roma[5], Miguel F. Segura[5‡], Olga Piskareva[1,2,3,7‡]*

1 Department of Anatomy and Regenerative Medicine, Cancer Bioengineering Group, RCSI University of Medicine and Health Sciences, Dublin, Ireland, 2 Department of Anatomy and Regenerative Medicine, Tissue Engineering Research Group (TERG), RCSI University of Medicine and Health Sciences, Dublin, Ireland, 3 School of Pharmacy and Biomolecular Sciences, RCSI University of Medicine and Health Sciences, Dublin, Ireland, 4 Faculty of Medicine, Cell Biology Unit, Department of Pathology and Experimental Therapeutics, University of Barcelona, Campus Bellvitge, Feixa Llarga s/n, L'Hospitalet de Llobregat, Spain, 5 Vall d'Hebron Research Institute, Group of Childhood Cancer & Blood Disorders, Universitat Autònoma de Barcelona, Barcelona, Spain, 6 Data Science Centre, School of Population Health, RCSI University of Medicine and Health Sciences, Dublin, Ireland, 7 Advanced Materials and Bioengineering Research Centre (AMBER), RCSI and TCD, Dublin, Ireland

‡ MFS and OP are Joint senior authors
* olgapiskareva@rcsi.com

**Data Availability Statement:** Data is contained within the article or Supporting Information

## Abstract

Neuroblastoma is the most common solid extracranial tumour in children. Despite major advances in available therapies, children with drug-resistant and/or recurrent neuroblastoma have a dismal outlook with 5-year survival rates of less than 20%. Therefore, tackling relapsed tumour biology by developing and characterising clinically relevant models is a priority in finding targetable vulnerability in neuroblastoma. Using matched cisplatin-sensitive KellyLuc and resistant KellyCis83Luc cell lines, we developed a cisplatin-resistant metastatic *MYCN*-amplified neuroblastoma model. The average number of metastases per mouse was significantly higher in the KellyCis83Luc group than in the KellyLuc group. The vast majority of sites were confirmed as having lymph node metastasis. Their stiffness characteristics of lymph node metastasis values were within the range reported for the patient samples. Targeted transcriptomic profiling of immuno-oncology genes identified tumour necrosis factor receptor superfamily member 4 (TNFRSF4) as a significantly dysregulated MYCN-independent gene. Importantly, differential *TNFRSF4* expression was identified in tumour cells rather than lymphocytes. Low *TNFRSF4* expression correlated with poor prognostic indicators in neuroblastoma, such as age at diagnosis, stage, and risk stratification and significantly associated with reduced probability of both event-free and overall survival in neuroblastoma. Therefore, *TNFRSF4* Low expression is an independent prognostic factor of survival in neuroblastoma.

material. RNA sequencing data "Neuroblastoma KellyLuc-Kelly83Luc xenografts, the HTG EdgeSeq immuno-oncology panel" is deposited at R2: Genomics Analysis and Visualization Platform (http://r2.amc.nl; ID: Xenograft Neuroblastoma KellyLuc (Immuno-oncology panel) - Piskareva - 10 - custom - rsg001).

**Funding:** O.P. received support for this project through Neuroblastoma UK, RCSI Strategic Academic Recruitment (StAR) Programme, Health Research Board - The Conor Foley Neuroblastoma Cancer Research Foundation (HRCI-HRB-2022-013). C.G. was funded by Irish Research Council Postgraduate Programme (GOIPG/2019/3220), R. S. - Irish Research Council - The Conor Foley Neuroblastoma Cancer Research Foundation (EPSPG/2021/95). M.S. was funded by Instituto de Salud Carlos III through the projects "ICI21/00076", "PI23/01144" and "PMP21/00073" (Co-funded by the European Regional Development Fund/European Social Fund; "A way to make Europe"/ "Investing in your future"). L.D.-J. is recipient of a Ramón y Cajal scheme (Grant No. RyC-2021-034346-I), funded by the Spanish Ministry for Science and Innovation (MCIN). The funders had no role in study design, data collection and analysis, decision to publish, or preparation of the manuscript.

**Competing interests:** The authors declare that they have no competing interests. The funders had no role in the design of the study; in the collection, analyses, or interpretation of data; in the writing of the manuscript, or in the decision to publish the results.

## Introduction

Neuroblastoma is a paediatric cancer of the sympathetic nervous system. It is the most common solid extracranial tumour in children and accounts for 15% of childhood cancer-related deaths [1–3]. Between one-third and half of all patients are classified as high risk and as such have an event-free survival (EFS) probability of less than 50%. This classification is based on clinical features such as an age at diagnosis, chromosomal alterations, amplification of the *MYCN* oncogene, which occurs in approximately 20% of neuroblastomas, and distant metastatic disease [4]. Implementation of intense multimodal therapy for neuroblastoma has increased the survival rate of patients with high-risk disease from 15% to nearly 50%. However, this regimen is still ineffective for approximately 50% of the high-risk cohort, resulting in relapse and the development of metastatic foci resistant to conventional therapies.

This rare nature of neuroblastoma restricts both the access to clinical material to study tumour biology and the number of clinical trials to assess new therapeutics when compared with adult cancers. Our understanding of its biology and response to treatment at the organism level is still limited, highlighting the demand for more clinically representative *in vitro* and *in vivo* models.

The tumour microenvironment (TME) in neuroblastoma is an area of growing interest in multiple areas, including risk stratification, prediction of drug response and targeted and immune therapies [5–12]. It is known that paediatric cancers in general have low immunogenicity, and this is no different for neuroblastoma, which displays a highly immunosuppressive microenvironment [5]. Insights into the molecular and proteomic changes in the TME of resistant neuroblastoma can be provided through the development of clinically and physiologically relevant therapy-resistant *in vitro* and *in vivo* models [13–17]. Our lab has successfully developed a panel of drug-resistant neuroblastoma cell lines using escalating exposure to cisplatin, followed by extensive characterisation of subsequent cell behaviour, cytotoxicity for several common chemodrugs, and genomic, miRNAomic and proteomic alterations [13]. In a follow-up study, we demonstrated that the cisplatin-resistant KellyCis83 cell line maintains its aggressive drug-resistant properties when grown both in 3D *in vitro* models and in an orthotopic murine model [14].

With the goal to move forward our research to another stage of the disease, this study aimed to characterise a cisplatin-resistant metastatic *MYCN*-amplified (MNA) neuroblastoma microenvironment by comparing the expression of a large panel of immuno-oncology genes using matched cisplatin-sensitive Kelly and resistant KellyCis83 cell lines injected into a metastatic growth model. We identified significantly dysregulated genes with clinical relevance in the cisplatin-resistant group and found genes with prognostic value and potentially targetable for novel therapy to treat high-risk neuroblastoma.

## Materials and methods

### Cell lines

Kelly (*MYCN* amplification, 17q chromosomal gain) and SK-N-AS (MYCN diploid, 1p and 11q chromosomal deletions, 17q chromosomal gain) were obtained from the European Collection of Authenticated Cell Cultures (ECACC) and are detailed in [18]. KellyCis83 was created by pulse exposure of Kelly cells to increasing concentrations of cisplatin over a 6-month period to a final cisplatin concentration of 83 μM and was extensively characterised by array comparative genomic hybridisation (aCGH), mass spectrometry, proliferation and toxicity assays[13]. KellyCis83 has 12 more chromosomal aberrations (9 gains, 2 losses, 1 homozygous deletion) compared to the parental Kelly. KellyCis83 doubling time is 33 hrs vs Kelly's 51 hr. Thus,

confirming that KellyCis83 as being both phenotypically and genomically distinct from the parental Kelly cell line. Kelly and KellyCis83 cell lines were transfected to contain a luciferase reporter gene encoded by pGL4.51 (luc2) (Promega) and are therefore referred to as KellyLuc and KellyCis83Luc, respectively. The SHEP-Tet-21N neuroblastoma cell line, which has been genetically modified to overexpress *MYCN*, where *MYCN* expression can be repressed by the addition of doxycycline to the culture media. SHEP-Tet-21N cells were obtained from Dr Louis Chesler with the permission of Prof. Manfred Swab [19].

## Murine metastatic neuroblastoma model

All animal experiments were performed at the Rodent Platform of the Laboratory Animal Service of the Vall d'Hebron Institute of Research (LAS, VHIR, Barcelona, Spain). All animal protocols were reviewed and approved by the regional Institutional Animal Care and Ethics Committee of Animal Experimentation (Ref. 39/20). All researchers participating in the experiment had animal experimentation certificate and had previous experience in similar procedures, ensuring proper handling of the animals and minimizing suffering or stress during their manipulation. Fox Chase SCID Beige mice, which have defective B cells, T cells and NK cells, were injected via tail vein with a suspension of either KellyLuc (cisplatin-sensitive, n = 10) or KellyCis83Luc (cisplatin-resistant, n = 10) cells. For the generation of metastasis, $2x10^6$ cells in 150μl of PBS were injected in the lateral tail vein of the mouse. Twenty animals were injected in total but one died during the injection procedure. This injection established metastatic disease in the mice, whose tumour burden was regularly examined using IVIS imaging via the luciferase reporter. For IVIS imaging purposes, mice were anesthetized in groups of no more than 5 animals using isoflurane. Once the anaesthesia's effect was confirmed, they were transferred to the imaging equipment where they will remain anesthetized through continuous infusion of isoflurane. All, but three xenografted mice were sacrificed on day 40 post-injection. Three of them were found dead due to undetermined reasons before meeting the endpoint criteria.

In order to minimize animal suffering, a protocol was established with strict criteria for monitoring weight loss, body condition, behaviour, physical appearance and growth of macro-metastases. Each criterion was scored from 0 to 3, being 0 n alteration and three the maximum alteration of the parameter. When a score of 3 was obtained in any category or a total of 8 points was combined across different parameters, the animal was euthanized in the next 0-12h after the score was determined. In the event of an increase in score compared to the immediately preceding supervision, and if the endpoint criteria are not met, the monitoring frequency was increase from 2 to 5 times/week. After reaching the ethical endpoint criteria, the mice were euthanized using $CO_2$ and cervical dislocation.

Metastatic foci were counted, formalin-fixed and paraffin embedded (FFPE) and stained with hematoxylin and eosin (H&E). This stain allowed the identification of tumour cell-enriched regions, indicated by dark purple staining, which were then macrodissected for RNA sequencing.

## RNA sequencing

RNA sequencing was carried out by ELDA Biotech using the HTG EdgeSeq human immuno-oncology panel. This assay examined the expression of 549 human immuno-oncology markers in cisplatin-sensitive and cisplatin-resistant xenografts, along with 15 housekeeping genes. Each gene was scored by log fold-change (LogFC) in KellyLuc vs. KellyCis83Luc xenografts, and *p* values were calculated via T test in Excel to identify significantly dysregulated genes. Cut-offs for selection of candidate genes were LogFC $>\pm0.6$ and $p\leq0.05$.

## Pathway enrichment analysis

Integrated differential expression and pathway analysis (iDEP) online software was used to apply hierarchical cluster analysis to the candidate gene panel [20]. This software ran k-means clustering, an unsupervised method for clustering genes into groups based on their expression pattern across all samples. It then identified the most significantly enriched Gene Ontology (GO) Biological Processes in the gene group. The LogFC and corrected $p$ values for each of the 36 candidate genes were input to achieve this.

Similarly, the Metascape gene annotation and analysis online platform was used to complement iDEP analyses, inputting the same LogFC and corrected $p$ values for each of the 36 genes [21]. This software assessed pathway and process enrichment using several ontology sources, such as Reactome, Gene Ontology (GO) Biological Processes and Kyoto Encyclopedia of Genes and Genomes (KEGG) Pathway.

## Kaplan–Meier survival analysis

To assess the clinical significance of each of the candidate genes, the R2 Genomics Analysis and Visualisation Platform (R2GAVP) was used [22] and IBM SPSS 21 software. This platform is a web-based microarray data repository for tumour data submitted by various European institutes and developed by the Department of Human Genetics in the Amsterdam Medical Centre (AMC). The SEQC neuroblastoma tumour cohort was selected, which contains gene expression profiles from 498 primary neuroblastomas using RNA-Seq and microarrays (S1 Table). The available identifiers for this cohort were sex, age at diagnosis, *MYCN* amplification status, risk status, INSS stage, disease course, tumour progression status and death from disease.

This dataset was used for Kaplan–Meier analysis by gene expression for each of the 36 candidate genes, generating survival curves for both overall survival (OS), defined as the length of time from either the date of diagnosis or the start of treatment for a disease, that patients diagnosed with the disease are still alive, and event-free survival (EFS), defined as the length of time after primary treatment for a cancer ends that the patient remains free of certain complications or events that the treatment was intended to prevent or delay. $p$ values were assessed for both survival outcomes, with a significance cut-off of $p \leq 0.05$.

## Assessment of candidate genes' clinical relevance

Analyses were run in R2GAVP to investigate relationships between the shortlisted genes and clinical features of neuroblastoma, including age at diagnosis, *MYCN* status, INSS stage, risk group, chromosomal alterations, disease course, tumour progression and death from disease. The SEQC cohort was used for this analysis, along with the Fischer cohort (S1 Table). The Fischer cohort was added to this analysis due to its available identifiers on chromosomal alterations, which are common in neuroblastoma and were not available for the SEQC dataset. To assess the potential regulation of shortlisted genes by the *MYCN* oncogene, gene correlation analyses were run between *MYCN* and each gene using the SEQC cohort. Y-Y plots with Log2 transformation were generated with R-values and $p$ values.

Low/High TNFRSF4 expression was generated based on Youden Index for the OS event. Correlation analysis between TNFRSF4 expression and the principal clinicopathological variables were performed (Fisher exact test and $X^2$). Then, univariate and multivariate Cox proportional hazard regression was performed to assess the predictive value of shortlisted genes for both EFS and OS, relative to other prognostic indicators including COG risk group, INSS stage, *MYCN*-amplification and age at diagnosis. This generated hazard ratios and $p$ values for

the likelihood of disease recurrence and death based on the expression levels of the selected genes. These statistical analyses were performed using the IBM SPSS 21 software.

## Analysing the cellular location of produced proteins

To target proteins that are primary participants in cell-to-cell interactions in the TME, we aimed to shortlist genes that produced proteins either expressed on the cell membrane or secreted from the cell. Each shortlisted gene from Kaplan–Meier analysis was examined using the Human Protein Atlas, focusing on the predicted location output. Genes producing proteins predicted to be secreted or expressed on the cell membrane were further shortlisted from this analysis.

## MYCN inducible system

To experimentally assess the influence of *MYCN* amplification on the expression of immuno-oncology genes, a *MYCN*-inducible system was used. SHEP-Tet21N cells were grown in the presence and absence of doxycycline to generate samples with and without *MYCN* expression. For *MYCN*-amplified samples (*MYCN*-on), SHEP-Tet21N cells were grown by conventional cell culture in RPMI-1640 media (Gibco #21875–034) supplemented with 10% foetal bovine serum (FBS) (Gibco #10270–106) and 1% penicillin–streptomycin (P/S) (Gibco #15140–122). For *MYCN*-repressed samples (*MYCN*-off), 1 µg/ml doxycycline (Sigma #17086-28-1) was added to the culture media, and the cells were grown in this medium for 6 days. After this time period, both sample types were taken down in QIAzol for RNA extraction.

## siRNA-mediated knockdown of *TNFRSF4*

To assess the impact of *TNFRSF4* expression on neuroblastoma cell survival, siRNA-mediated gene silencing was carried out on the SKANS and KellyLuc cell lines. Cells were plated in 96-well tissue culture plates at $2x10^4$ cells/well and were allowed to adhere and grow at 37˚C and 5% $CO_2$ for 24 h. Cells were then transfected with either siTNFRSF4 (Invitrogen #AM16708), siKIF11 (Invitrogen #4390824) or a scrambled negative control (Ambion #AM4611). On the morning of transfection, conditioned growth media was replaced with 100 µl of fresh media without FBS or P/S. For each well to be transfected, 0.3 µl of siRNA/negative control was mixed with 5 µl of Opti-MEM™ Reduced Serum Medium (Gibco #31985–070). Similarly, for each well, 0.1µl of Lipofectamine™ 3000 Transfection Reagent (Invitrogen #L3000-008) was mixed with 5µl of Opti-MEM™ Reduced Serum Medium. These two solutions were mixed together and incubated at room temperature for 20–30 min to allow complexes to form. The siRNA duplex-Lipofectamine™ 3000 solution was then added to appropriate wells, and the plate was gently rocked back and forth to mix before being placed in the tissue culture incubator. After 4 h, 35 µl of complete media was added to each well, and the plates were returned to the incubator. After 72 h, cellular proliferation was assessed via DNA quantification using the Quant-iT PicoGreen dsDNA Assay Kit (Invitrogen #P11496), and cell viability was assessed using the CytoTox-Glo™ Assay (Promega, #G9290). Samples were also taken down in QIAzol for RNA extraction.

## RNA extraction

Total RNA was extracted from fresh-frozen xenograft tissue for gene expression analysis via reverse transcription quantitative PCR (RT–qPCR). Fresh-frozen xenograft tissue that matched the FFPE samples was added to 700 µl of QIAzol Lysis Reagent (QIAGEN #79306), and samples were then disrupted using Tissue Lyser LT (QIAGEN #85600) for 2–4 min at 50

oscillations/second. The QIAGEN miRNeasy kit (QIAGEN #217084) was then used to purify total RNA from all samples. Isolated RNA was quantified, and its purity was assessed by A260/A280 and A260/230 ratios, calculated on a NanoDrop™ (Thermo Fisher Scientific #ND2000).

## RT–qPCR

Reverse transcription (RT) was used to generate complementary DNA (cDNA) from the extracted xenograft RNA. This was performed using the High-Capacity cDNA Reverse Transcription Kits (Applied Biosystems #4374966) as per the kit's guidelines in 0.2 ml PCR Eppendorf tubes (Sigma–Aldrich #EP0030124359). The RT reaction was run on a Veriti 96-well thermal cycler (Applied Biosystems #4375305) under the following conditions: 25˚C (10 min), 37˚C (120 min), 85˚C (5 min) and 4˚C ($\infty$). The resulting cDNA samples were stored at -20˚C until ready for quantitative PCR (qPCR).

For qPCR analyses, predesigned TaqMan assays were obtained from Applied Biosystems (#4331182) for the following genes: *TNFRSF4* (Assay ID HS02559990_s1), *TNFRSF1B* (Assay ID HS00153550_m1), *GNLY* (Assay ID HS01120098_g1), *SOCS3* (Assay ID HS02330328_s1) and *GAPDH* (Assay ID HS02786624_g1). Reactions were carried out in MicroAmp™ Optical 96-Well Reaction Plates (Applied Biosystems #N8010560) with each sample assayed in technical triplicate. Reactions were run on the 7500 Real-Time PCR System (Applied Biosystems #4351105) under the following conditions: 50˚C (2 min), 95˚C (10 min), [95˚C (15 sec), 60˚C (1 min)] x 40 cycles.

Cycle threshold (Ct) values were obtained from the 7500 system, and a Ct value >35 was taken as no detection. Expression was normalised using the global average method, where the average Ct value of all targets across all samples was subtracted from the Ct values for any given sample and target ($\Delta$Ct). The $\Delta$Ct of the KellyLuc xenograft samples was then subtracted from the $\Delta$Ct of the KellyCis83Luc xenograft samples ($\Delta\Delta$Ct). Finally, the relative quantification value (RQ) was calculated by raising 2 to the power of minus the $\Delta\Delta$Ct of each sample ($2^{-\Delta\Delta Ct}$).

## Mechanical testing

Mechanical testing was performed as previously described [23]. Prior to testing, KellyLuc and KellyLucCis83 tumours were thawed in ice-cold phosphate-buffered saline (PBS) for 1 hour. Once thawed, the width, length and height of tumours were measured using digital callipers. Tumours were also weighed before being placed back on ice. The mechanical properties of tumours were probed using a mechanical testing machine (Zwick/Roell, Ulm, Germany) fitted with a 5 N load cell. Uniaxial, unconfined compression tests were performed in an ice-cold PBS bath between impermeable platens. Tumours were compressed at a strain rate of 10%/minute to a final strain of 10%. The stress experienced by each sample was calculated by dividing the applied force by the cross-sectional area of the tumour, and the resulting stress–strain curve was plotted. The compressive modulus of each tumour was defined as the slope of a linear fit to the stress–strain curve between 2% and 5% strain.

## Results

### Cisplatin-resistant neuroblastoma cells create more aggressive disease

To shed light on the contribution of cisplatin-resistant *MYCN*-amplified neuroblastoma cells in the tumour immune microenvironment, we established a murine xenograft model using the cisplatin-sensitive cell line KellyLuc and its cisplatin-resistant derivative cell line KellyCis83Luc [13], which were injected into the tail vein of mice and allowed to metastasise for 40 days before resection (Fig 1, S1 and S2 Figs). Resected tumours were then assessed by a

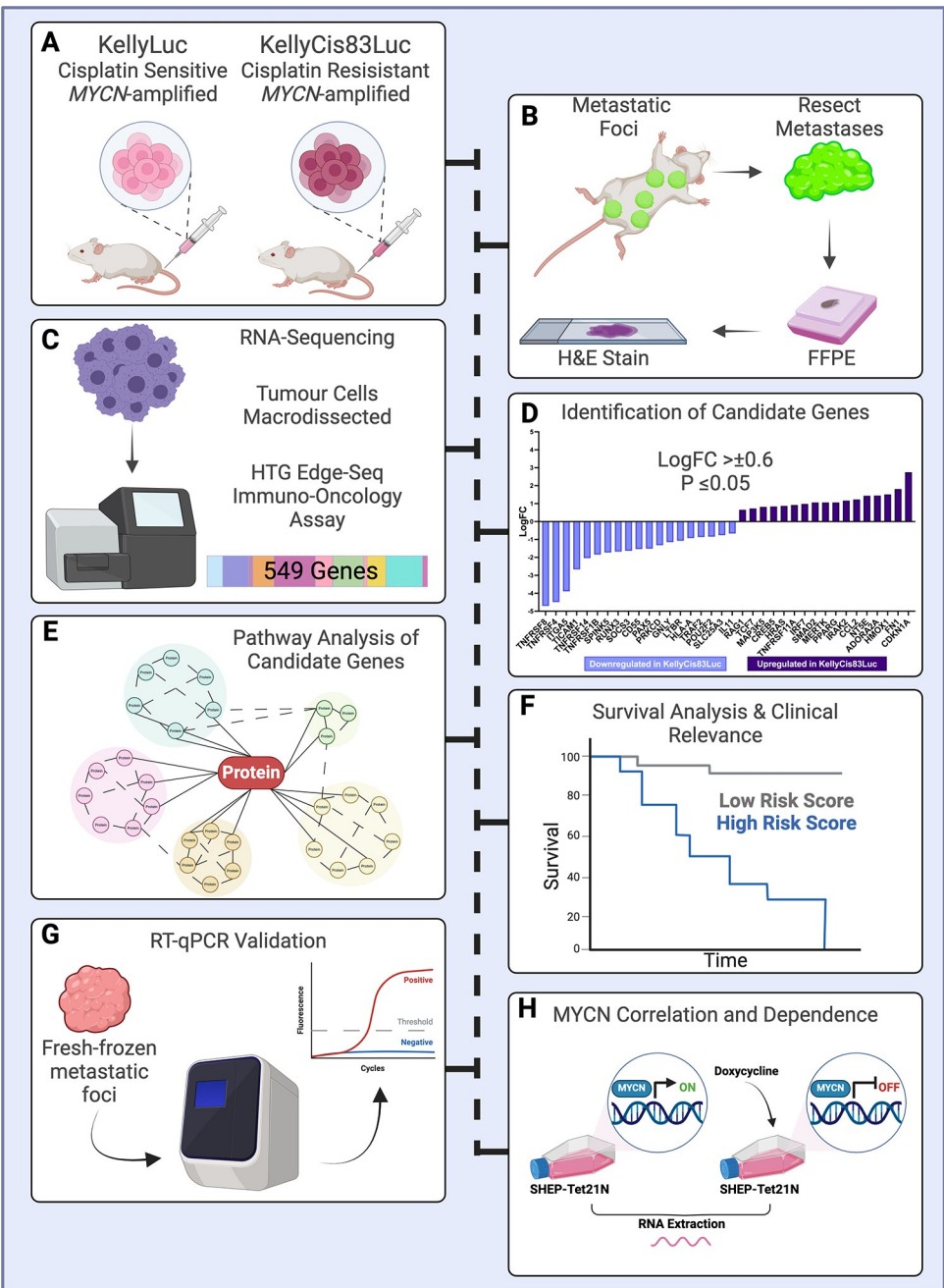

**Fig 1. Cisplatin-resistant KellyCis83Luc tail-vein injections cause more metastases in murine xenografts than drug-sensitive KellyLuc injections.** (A) Mean number of metastases per mouse for both cell lines. (B) Percentage of mice with more than 10 metastatic foci per cell line. (C) Metastatic foci grown in immunodeficient mice injected with the KellyLuc and KellyCis83Luc cell lines by tail vein. The graph represents the number and location of metastases in the indicated cell lines. (D) Schematic of major murine lymph nodes. (E) Representative histology of metastatic foci in each location by cell line. Scale bars: 1000 μm for whole slide images and 100 μm for magnifications. (F) Stiffness of lymph node metastasis tissues of the two cell lines KellyLuc (n = 2) and KellyCis83Luc (n = 5). Mets–metastases, Kelly–KellyLuc, KellyCis–KellyCis83Luc.

pathologist to verify the presence of cancerous cells. Tumour cell-enriched regions were then assessed via transcriptomic profiling to identify significantly dysregulated immuno-oncology markers in cisplatin-resistant tumours (*S3 Fig*).

Cisplatin-resistant KellyCis83Luc xenografts demonstrated more aggressive metastatic behaviour, with a total of 103 metastatic foci resected from the group of 9 mice, compared to 57 in the cisplatin-sensitive control group (Fig 1A and 1B). Metastases were identified in many sites, including the kidney, liver, abdomen, thorax, pelvis, axillary artery, maxilla and cervical spine (Fig 1C). The vast majority of metastases were confirmed as lymph node metastasis (Fig 1D and 1E). The average number of metastases per mouse was significantly higher in the cisplatin-resistant model than in the cisplatin-sensitive control group (11.4 vs 6.3, $p = 0.019$), and there was a larger percentage of mice with more than 10 metastases in the cisplatin-resistant group (66.7 vs 11.1).

Additionally, we biomechanically characterised xenografts (Fig 1F). The lymph node metastases of xenografts analysed here showed stiffnesses between 2.4671 kPa and 14.6880 kPa.

## The expression of immuno-oncology genes is altered in cisplatin-resistant neuroblastoma

Resected FFPE metastatic foci were stained with H&E (S3 Fig) to identify tumour cell-enriched regions that were macrodissected for RNA sequencing via the HTG EdgeSeq Immuno-Oncology Assay, which profiles the expression of 549 genes. This analysis identified a total of 36 candidate genes that were dysregulated in cisplatin-resistant xenografts, 19 of which were significantly downregulated compared to the cisplatin-sensitive control group, and 17 of which were significantly upregulated (LogFC $>\pm0.6$ and $p \leq 0.05$) (Fig 2).

## Functional enrichment analysis identifies pathways modulating lymph node metastasis formation

Pathway enrichment analysis via iDEP software was used to gain insights into the molecular pathways underlying gene expression changes observed in cisplatin-resistant metastases. iDEP software sorted the 36 candidate genes into 4 distinct clusters (Fig 3A), with clusters A and B containing genes that were upregulated in cisplatin-resistant KellyCis83Luc metastatic foci and clusters C and D containing genes that were downregulated. This analysis was verified by Metascape online software, which uses a number of ontology sources, including Reactome, GO Biological Processes and KEGG Pathways, to identify and plot the top 20 enriched terms from the input genes (S4 Fig).

iDEP Cluster A had the second highest number of enriched pathways or processes. This cluster contained 13 (36%) of the candidate genes (*HMOX1*, *CCL2*, *PPARG*, *MAP3K5*, *HRAS*, *IRF7*, *ADORA2A*, *IRAK2*, *NT5E*, *TNFRSF11A*, *MERTK*, *RAG1*, *SMAD2*) assigned to the cell surface receptor signaling pathway ($p = 2.28\text{e-}05$) and cellular response to cytokine stimulus ($p = 2.28\text{e-}05$), which are known to be involved in modulation of the lymph node microenvironment during tumour metastasis. The prediction agreed with the Metascape outputs, which highlighted the enriched processes of cytokine signaling in the immune system ($p = 6.3\text{e-}22$, gene count [N] = 19), regulation of cell activation ($p = 9.4\text{e-}19$, N = 19), and regulation of molecular mediator of immune response ($p = 3.22\text{e-}13$, N = 17). Six of the candidate genes were common to the three most enriched processes: *HLA-A*, *HMOX1*, *CCL2*, *TNFRSF1B*, *TNFRSF4* and *TNFRSF14*.

Interestingly, *TNFRSF1B*, *TNFRSF4* and *TNFRSF14* were assigned by iDEP to Cluster D with the highest number of enriched pathways or processes (N = 16). This cluster contained 15 (42%) of the candidate genes (*TNFRSF14*, *CD55*, *HLA-A*, *TNFRSF1B*, *POU2F2*, *TRAF2*, *TICAM1*, *SPINK5*, *RUNX3*, *GNLY*, *PRKCD*, *LTBR*, *IL11*, *SOCS3* and *PAX5*) and contained the

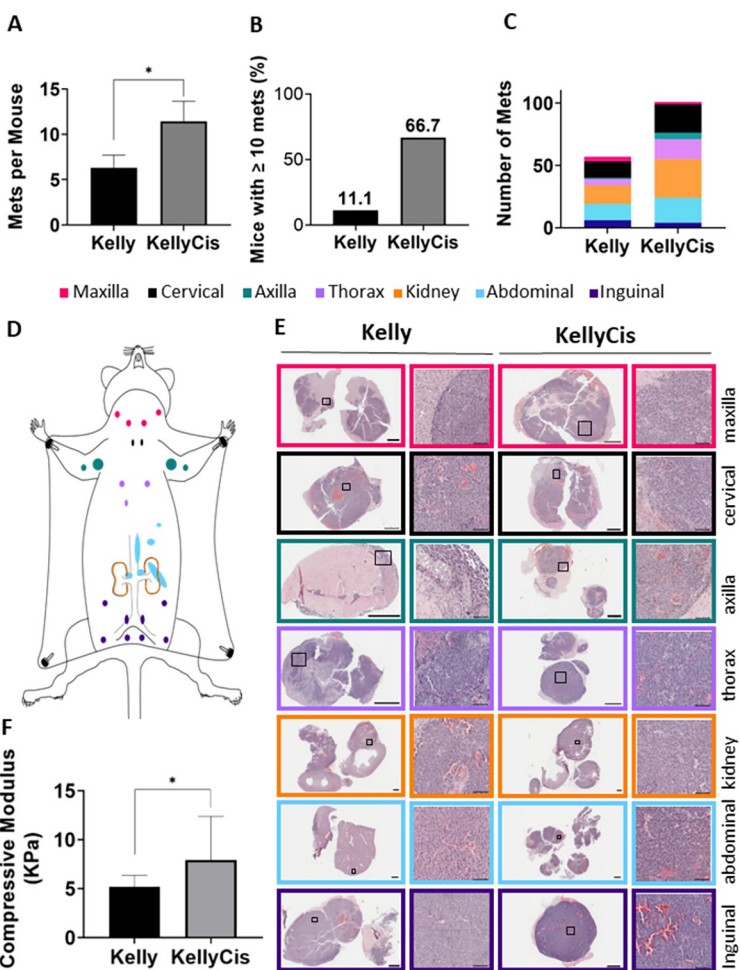

**Fig 2. Establishment and analysis of a drug-resistant neuroblastoma murine xenograft model.** A) Fox Chase SCID Beige mice were injected with either cisplatin-sensitive KellyLuc or cisplatin-resistant KellyCis83Luc cells. B) After 40 days, metastases were resected, formalin-fixed paraffin embedded (FFPE) and stained with hematoxylin and eosin (H&E) to identify tumour cell-enriched regions. C) These positive regions were macrodissected for RNA sequencing via the HTG EdgeSeq Immuno-Oncology assay. D) The log fold-change (logFC) in the expression of each gene in cisplatin-sensitive vs cisplatin-resistant tumours was assessed. Using cut-off values of LogFC >±0.6 and $p \leq 0.05$, 36 candidate genes were identified. E) Pathway analysis software was used to gain insights into the molecular pathways underlying the candidate gene expression patterns. F) Kaplan–Meier survival analysis was conducted to determine the clinical relevance of candidate genes in neuroblastoma. G) Expression trends of shortlisted genes were validated by RT–qPCR on fresh-frozen xenograft tissue. H) The correlation between shortlisted genes and *MYCN* was calculated, and *MYCN* dependence was assessed via an inducible system. Created with biorender.com.

most significantly enriched processes: immune system process ($p$ = 3e-09), lymphocyte activation ($p$ = 3e-09) and production of molecular mediator of immune response ($p$ = 2e-09).

## Immune-oncology related genes are associated with neuroblastoma outcomes

Following pathway enrichment analysis, the clinical relevance of the 36 immuno-oncology candidate genes in neuroblastoma was assessed in a large cohort SEQC of 498 neuroblastomas (S5 Fig) [24] [ref]. The genes' impact on both overall survival (OS) and event-free survival (EFS) was assessed, and $p$ values were used to determine whether the candidate genes were clinically significant. To identify those genes that could functionally contribute to

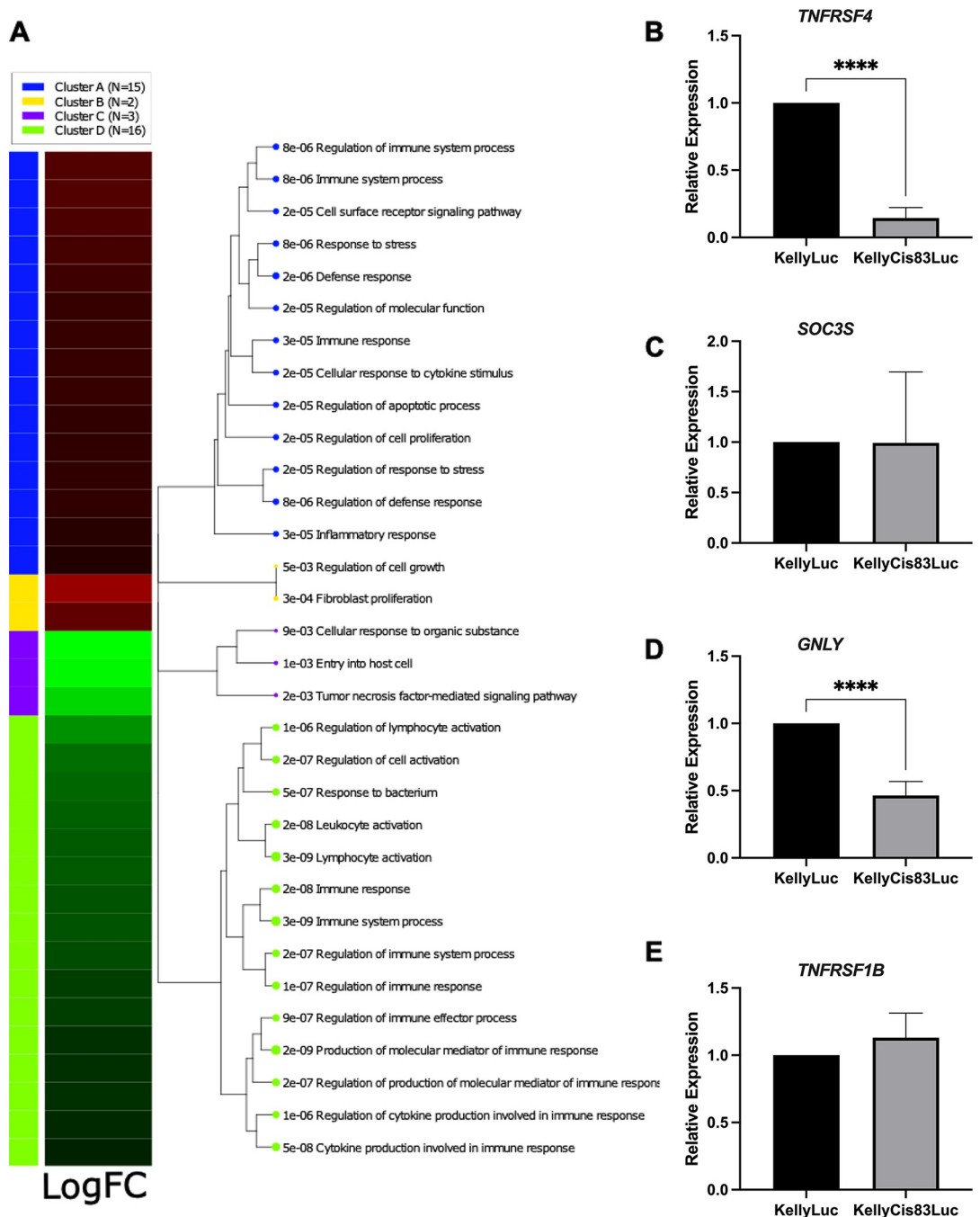

**Fig 3. Functional enrichment analysis identifies pathways of immune system-related processes.** A) iDEP software analyses. N refers to the number of enriched GO biological processes for each cluster. For each enriched process, a *p* value is given. Gene sets closer on the tree share more genes. Dot sizes correspond to adjusted *p* values, the higher p value the bigger the dot. (B-E) Validation of some differentially expressed genes by RT–qPCR.

neuroblastoma aggressiveness, we assessed whether the Kaplan–Meier expression trends matched the expression trends observed in the RNA-seq data. Twelve genes were shortlisted from the panel of 36 candidate markers. Seven genes were significant for both OS and EFS: *TNFRSF4*, *TNFRSF1B*, *TNFRSF14*, *GNLY*, *PAX5*, *PRKCD* and *RUNX3*, and five genes were

significant for OS only: *ITGA5*, *HLA-A*, *HRAS*, *POU2F2* and *SOCS3* (S5 Fig). Eleven of the shortlisted genes were downregulated in cisplatin-resistant tumours, low expression was found to be unfavourable in neuroblastoma, one gene, *HRAS*, was upregulated in cisplatin-resistant tumours, and high expression was found to be unfavourable in neuroblastoma.

Because a direct neuroblastoma–immune cell interaction would be facilitated by either proteins secreted or expressed on the cancer cell surface, we selected *TNFRSF4*, *TNFRSF1B*, *GNLY* and *SOCS3* for validation of expression trends via RT–qPCR in fresh-frozen cisplatin-sensitive and resistant metastatic tissue matched to the FFPE tissue used in RNA-seq. Such validation is warranted when evaluating RNA-seq based expression profiles [25]. *TNFRSF4* (Fig 3B) and *GNLY* (Fig 3D) were found to be significantly downregulated in cisplatin-resistant xenografts (*p*<0.0001). This agrees with trends observed in RNA-seq data where *TNFRSF4* was significantly downregulated with a LogFC of -4.5 (*p* = 1.36e-04) and *GNLY* was significantly downregulated with a LogFC of -1.15 (*p* = 7.95e-03). However, there was no significant difference in the expression of *SOCS3* (Fig 3C) or *TNFRSF1B* (Fig 3E) in cisplatin-sensitive vs resistant tumours through this analysis. A decreased *TNFRSF4* expression was additionally verified at the genomic and transcriptomics levels in KellyLuc and KellyCis83Luc cell lines (S6 Fig). As *TNFRSF4* displayed greater significance in Kaplan–Meier survival analysis and because this marker has been researched more extensively in cancers outside of neuroblastoma, *TNFRSF4* was selected for further studies.

## Low *TNFRSF4* expression is correlated with poor prognostic indicators in neuroblastoma

The expression of *TNFRSF4*, which encodes the protein OX40, was assessed across different patient subgroups in large neuroblastoma cohorts to determine whether its expression could have prognostic value (Fig 4, S1 Table). In terms of clinical outcomes, we observed that Low expression of *TNFRSF4* significantly reduced survival probabilities in neuroblastoma in the SEQC cohort of 498 neuroblastomas in terms of both OS (*p* = 1.53e-09) and EFS (*p* = 4.2e-04) (Fig 4A and 4B). This was strengthened by the assessment of patient subgroups in terms of adverse clinical outcomes in the SEQC cohort, where *TNFRSF4* was also significantly lower in patients with an unfavorable disease course (N = 91, patients died despite intensive chemotherapy) than in patients with a favourable disease course (N = 181, patients survived without chemotherapy for at least 1000 days postdiagnosis). Patients with a progression event such as metastasis or relapse also exhibited significantly lower *TNFRSF4* expression (N = 183), and finally, TNFRSF4 expression was significantly lower in patients who died from neuroblastoma (N = 105) (Fig 4C). These results collectively demonstrate that underexpression of *TNFRSF4*, as observed in our cisplatin-resistance model, is associated with adverse clinical outcomes in neuroblastoma as well as poor prognostic indicators. In the SEQC cohort, *TNFRSF4* expression was found to be significantly reduced in patients with *MYCN* amplification (N = 92). The *MYCN* oncogene is amplified in approximately 20% of neuroblastomas and is the most well-defined oncogenic driver and indicator of poor prognosis in neuroblastoma, an essential feature for risk group stratification in neuroblastoma. Cohn et al found that *MYCN* amplification can reduce EFS and OS by almost 50% (87% non-MNA vs 46% MNA and 95% non-MNA vs 53% MNA, respectively). *TNFRSF4* expression was also significantly lower in patients classified as high-risk (N = 176) (Fig 4C). Risk stratification is multifactorial, considering the International Neuroblastoma Risk Group (INRG) tumour stage, patient age at diagnosis, tumour histology, differentiation, *MYCN* status, DNA ploidy and chromosomal imbalances. It is estimated that approximately 36% of neuroblastoma patients are high-risk, with survival rates of <50%. Patient age at diagnosis was also found to be associated with the expression of

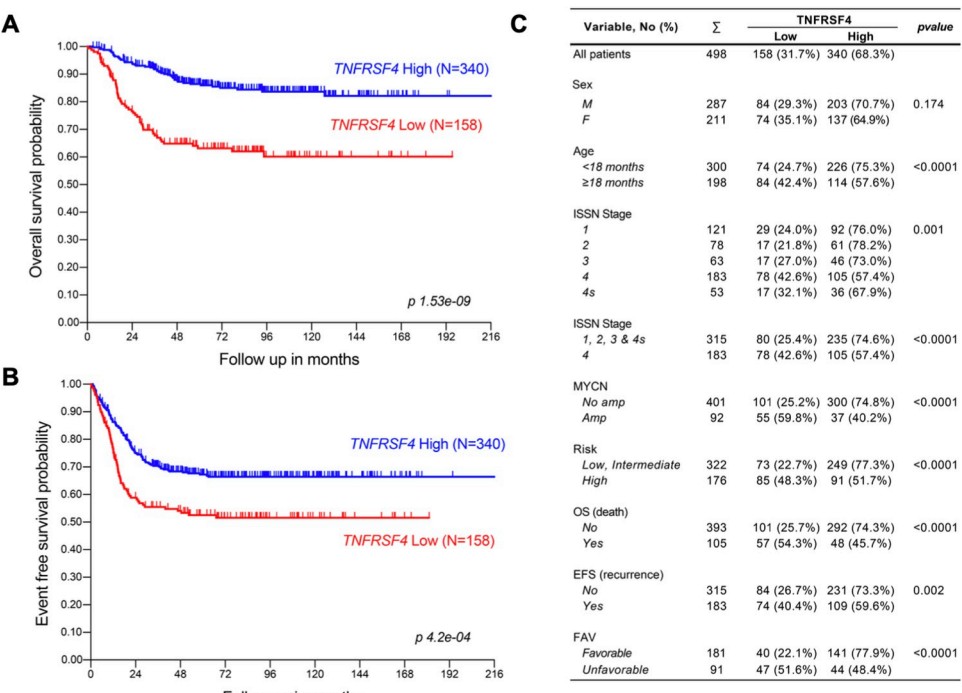

**Fig 4. Clinical relevance of *TNFRSF4* in neuroblastoma.** Kaplan-Meier analysis of *TNFRSF4* expression for OS (A), EFS (B), *TNFRSF4* mRNA expression correlations with clinical variables in neuroblastoma (N = 498) (C) EFS: Event-Free Survival; FAV: unfavorable/favorable (class label for extreme disease course); HR: High-Risk patients; OS: Overall Survival. High-Risk: patients with stage 4 disease >18 months at diagnosis and patients of any age and stage with *MYCN*-amplified tumours. The data generated using the IBM SPSS 21 software.

*TNFRSF4*, with significantly lower expression in patients older than 18 months at diagnosis (N = 198) (Fig 4C), with patients younger than this at diagnosis having a much better prognosis. The SEQC dataset did not provide any information on chromosomal deletions or insertions; however, the Fischer cohort of 223 neuroblastomas (S1 Table) did provide this information and demonstrated that *TNFRSF4* expression was significantly lower in patients with a chromosome 1p deletion (N = 67, *p* = 9.71e-03). Deletion or loss of heterozygosity (LOH) at chromosome 1p36 is observed in approximately 23% of neuroblastomas and is strongly associated with *MYCN* amplification and decreased EFS. The *TNFRSF4* gene is located at chromosome 1p36.33, which concurs with reduced expression in those with 1p36 LOH. Data generated by the Therapeutically Applicable Research to Generate Effective Treatments (TARGET) initiative phs000467 also found that *TNFRSF4* was homogenously deleted in 5% of neuroblastoma cases. Taken together, these results clearly demonstrate that the low expression of *TNFRSF4* observed in our cisplatin-resistant xenografts is significantly associated with poor prognostic indicators of neuroblastoma.

We next assessed the predictive ability of *TNFRSF4* for patient survival in neuroblastoma using both univariate and multivariate Cox regression models. For univariate analysis, the hazard ratios for recurrence and death were calculated for the COG risk group (high-risk vs low/intermediate-risk), INSS stage (stage 4 vs stages 1/2/3/4S), *MYCN* amplification (MNA vs non-MNA), age at diagnosis (≥18 months vs <18 months), and finally expression of *TNFRSF4* (Low vs High) (Fig 5). As expected, the risk group gave rise to high hazard ratios, where high-risk patients were 5.2 times more likely to experience recurrence and 21.4 times more likely to die. INSS stage 4 patients were almost 3.9 times more likely to experience recurrence and 8.7

| Variable | EFS (recurrence) | | OS (death) | |
|---|---|---|---|---|
| | Hazard Ratio (95% CI) | *p* | Hazard Ratio (95% CI) | *p* |
| **Risk Group** High vs Low/Intermed | 5.180 (3.800-7.061) | <0.0001 | 21.423 (11.932-38.464) | <0.0001 |
| **INSS Stage** 4 vs 1/2/3/4S | 3.891 (2.875-5.266) | <0.0001 | 8.660 (5.441-13.783) | <0.0001 |
| *MYCN* MNA vs non-MNA | 3.217 (2.349-4.405) | <0.0001 | 7.793 (5.262-11.542) | <0.0001 |
| **Age** ≥18m vs <18m | 3.053 (2.259-4.126) | <0.0001 | 8.114 (4.980-13.221) | <0.0001 |
| *TNFRSF4* Low vs High | 1.692 (1.259-2.274) | <0.0001 | 3.081 (2.097-4.525) | <0.0001 |

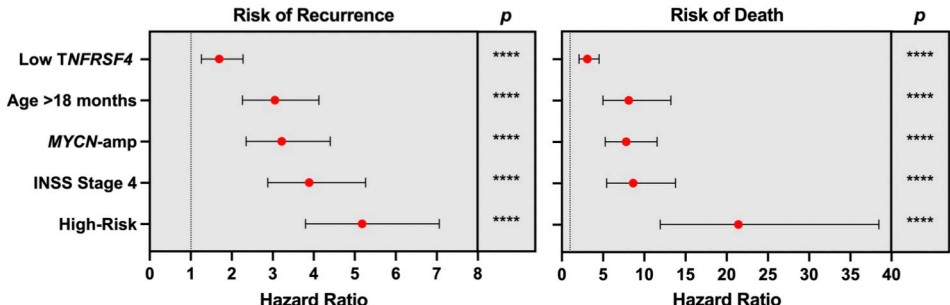

**Fig 5. Univariate Cox regression analysis.** The SEQC neuroblastoma cohort was used for univariate Cox regression analysis for multiple prognostic variables in neuroblastoma, including the expression of *TNFRSF4*. Forest plots were generated based on the hazard ratios, 95% confidence intervals (CIs) and significance values (*p*). The data generated using the IBM SPSS 21 software.

times more likely to die. *MYCN* amplification increased the likelihood of recurrence 3.2-fold and death 7.9-fold, and age at diagnosis over 18 months similarly increased the likelihood of recurrence 3-fold and death 8.1-fold. Finally, patients with low expression of *TNFRSF4* were 1.7 times more likely to experience recurrence and 3.1 times more likely to die.

To ensure that each variable was significant independent of the others, multivariate Cox regression was also performed for INSS stage, *MYCN* amplification, age at diagnosis and *TNFRSF4* expression (Fig 6). COG risk group was excluded from this analysis, since multivariate analysis should include individual clinical variables independent from each other. Hazard ratios were reduced for each variable for both recurrence and death using the multivariate model; however, INSS Stage 4 disease, *MYCN* amplification and age >18 months at diagnosis still significantly increased the likelihood of both recurrence and death. The increased likelihood of recurrence based on Low expression of *TNFRSF4* was no longer significant in this analysis, but this variable remained significant at increasing the likelihood of death 1.7-fold (*p* = 0.007). Then, *TNFRSF4* Low expression was an independent prognostic factor of survival in neuroblastoma.

## *MYCN* is not a direct repressor of *TNFRSF4*

Owing to the significant inverse correlation shown between *MYCN* and *TNFRSF4* in multiple neuroblastoma datasets (Figs 4–6 and S7 and S8 Figs), we next aimed to examine whether *TNFRSF4* could be a bonafide MYCN target. For that purpose, we took advantage of the SHEP-Tet-21N *MYCN*-inducible cell line. In this system, the expression of *MYCN* can be

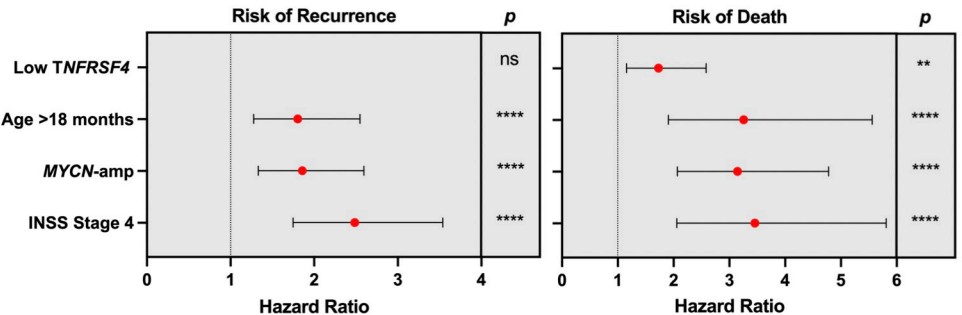

| Variable | EFS (recurrence) | | OS (death) | |
|---|---|---|---|---|
| | Hazard Ratio (95% CI) | p | Hazard Ratio (95% CI) | p |
| **INSS Stage** 4 vs 1/2/3/4S | 2.486 (1.747-3.539) | <0.0001 | 3.457 (2.057-5.810) | <0.0001 |
| *MYCN* MNA vs non-MNA | 1.858 (1.331-2.594) | <0.0001 | 3.144 (2.068-4.780) | <0.0001 |
| **Age** ≥18m vs <18m | 1.802 (1.276-2.547) | 0.001 | 3.256 (1.907-5.560) | <0.0001 |
| *TNFRSF4* Low vs High | - | 0.211 | 1.730 (1.158-2.584) | 0.007 |

**Fig 6. Multivariate Cox regression analysis.** The SEQC neuroblastoma cohort was used for multivariate Cox regression analysis to assess whether the hazard ratios for INSS stage 4, *MYCN* amplification, age at diagnosis ≥ 18 months and low expression of *TNFRSF4* were significant independent of each other. Forest plots were generated based on the hazard ratios, 95% confidence intervals (CI) and significance values (*p*). The data generated using the IBM SPSS 21 software.

repressed through the addition of doxycycline to the growth media (Fig 7B). We collected RNA from SHEP-Tet21N cells grown both in the presence and absence of doxycycline and assessed *TNFRSF4* expression via RT–qPCR (Fig 7C). The addition of doxycycline resulted in 20-fold lower expression of *MYCN* in SHEP-Tet21N cells (p<0.0001), showing that the oncogene was successfully repressed. However, there was no significant difference in the expression of *TNFRSF4* observed (*p* = 0.721), thereby suggesting that TNFRSF4 expression is *MYCN*-independent (Fig 7C).

## *TNFRSF4* knockdown minimally inhibits cell viability

To assess whether *TNFRSF4* expression was functionally relevant important for the oncogenic capacities of neuroblastoma cells, we carried out siRNA-mediated knockdown of *TNFRSF4* in Kelly (*MYCN* amplified) and SK-N-AS (non-*MYCN* amplified) neuroblastoma cells grown *in vitro* (Fig 7D). Cells were transfected with either a silencer select negative control (NC), siRNA against *TNFRSF4* (siTNFRSF4) or against the kinesin-like protein KIF11 (siKIF11), a molecular motor protein that plays an essential role in mitosis and can serve as a positive control for cell viability and proliferation [26]. Cellular viability was assessed 96 h posttransfection using the CytoTox-Glo™ Assay. The knockdown of *TNFRSF4* did not affect the viability of Kelly cells and slightly reduced the viability of SK-N-AS cells compared to NC (~12%, *p* = 0.03). RT–qPCR results confirmed significant downregulation of *TNFRSF4* in samples transfected with siTNFRSF4 compared to NC (p<0.0001) (Fig 7E). These results suggest that TNFRSF4 is dispensable for the viability of neuroblastoma cells. However, its implication in other metastatic-related processes such as adhesion, migration or invasion remains to be elucidated.

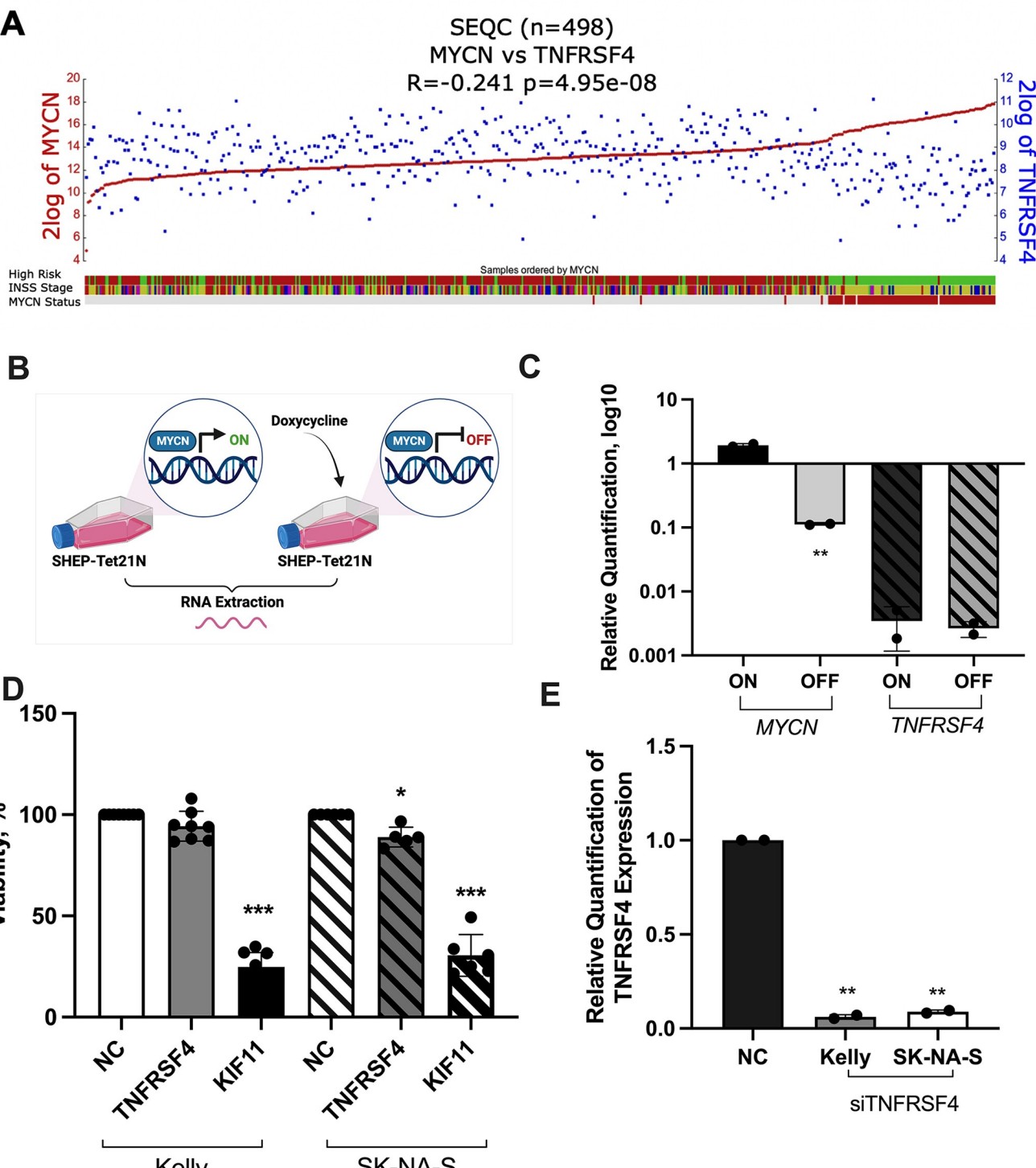

**Fig 7. Assessment of *TNFRSF4* regulation by the *MYCN* oncogene and impact on cellular proliferation.** A) Correlation between *MYCN* and TNFRSF4 expression in a cohort of 498 neuroblastomas generated in R2. B) SHEP-Tet-21N cells were grown in the presence (*MYCN*-off) and absence (*MYCN*-on) of doxycycline, and RNA was extracted for RT–qPCR to assess whether the expression of *TNFRSF4* is *MYCN* dependent. C) Expression of *MYCN* and *TNFRSF4* in SHEP-Tet21N cells grown in the presence and absence of doxycycline. D) Kelly and SK-N-AS cells were transfected with either a scrambled negative control (NC), siTNFRSF4 or siKIF11. Cell viability was assessed using the CytoTox-Glo™ Assay. The values were converted to a percentage of the viable cells compared to the negative control. The results of 2 independent experiments run in 3 biological replicates were combined. These data were statistically analysed using one-way ANOVA to determine whether cell viability was significantly reduced as a result of transfection. E) RT–qPCR validation of downregulation of TNFRSF4 expression in Kelly and SK-N-AS cell transfectants. Two-tailed unpaired T tests were performed in GraphPad Prism to detect significant differences in

expression (*p $\leq$ 0.05; **p $\leq$ 0.01; ***p$\leq$ 0.001; ****p $\leq$ 0.0001). Analysis confirmed significant downregulation of TNFRSF4 in cells transfected with siTNFRSF4 compared to NC. Image created with Biorender.com.

## Discussion

Here, we described a cisplatin-resistant metastatic neuroblastoma model established by the tail vain injection of cisplatin-sensitive Kelly and its cisplatin-resistant derivative KellyCis83 neuroblastoma cell lines [13]. Our study identified a potential relationship between the drug-resistant phenotype, which causes relapse in approximately 50% of neuroblastomas, the formation of a sentinel lymph node microenvironment during tumour metastasis and immune-related gene expression alterations [2].

While ideally preclinical models for studying cancer should be immunocompetent, the options for neuroblastoma are limited. The described murine model host was Fox Chase SCID Beige mice, which are absent in mature B cells and T cells and have defective NKs; however, DCs and macrophages are present, and the complement system is functional.

In the described metastatic model, both Kelly and KellyCis83 cells disseminated to a variety of lymph nodes at least partially by spontaneous lymphatic dissemination of cancer cells.

One of the most intriguing observations is an almost twofold increase in metastatic aggressiveness of cisplatin-resistant KellyCis83Luc mets compared to the cisplatin-sensitive control group. While an increase in the number of detectable metastases of KellyCis83 could be explained by increased doubling time [13] and alterations in the cell cycle [27], their invasion potential *in vivo* does not correlate with the same *in vitro*. The initial assessment of the migration and invasion potential of Kelly and KellyCis83 cells revealed no significant change *in vitro* [13]. We speculate that the metastatic potential could also be influenced by host micro- and macro- environmental factors after cell injection. One such environmental factor is tumour stiffness, a biomechanical property that is well known to be altered in tumours, and typically stiffer tissues are linked to more severe tumours [28,29]. Using polyacrylamide gel models of varying stiffnesses, Lam et al determined that in neuroblastoma, the opposite appears to be the case. Stiffer models promoted greater differentiation, reduced MYCN expression and less proliferation [30]. Another study found a bimodal distribution ranging from soft samples of 0.174 kPa to the stiffest sample at 8.452 kPa for patient samples of neuroblastoma [31]. We observed that lymph node metastasis values were within the range reported for the patient samples, although on the stiffer side. This may be due to differences in tissue of origin or severity of the tumours; the latter was not investigated in relation to stiffness.

Pathway enrichment analysis of the profiled immuno-oncology panel via iDEP and Metascape points towards some dysregulated genes that can modulate the formation of metastatic sites through cytokine signalling and the production of other molecular mediators of the immune response in the TME. One of the cytokines identified was CCL2 (C-C motif chemokine ligand 2), which was upregulated in cisplatin-resistant tumours (logFC = 1.23, $p$ = 0.011, S9 Fig). CCL2 belongs to a group of low molecular weight cytokines with chemoattractant activity that binds the cognate receptor CCR2. The CCL2-CCR2 pair has a long track record of multiple protumorigenic roles, including the promotion of tumour growth and metastasis in many cancer types [32]. Elevated levels of CCL2 increased tumorigenicity and promoted metastasis to the lymph nodes in gastric cancer, bladder cancers and melanoma [33,34]. While CCL2 expression by tumour cells can stimulate nontumour cells via a paracrine mechanism, CCL2 can also activate tumour cells via an autocrine mechanism [35]. Thus, the increased number of cisplatin-resistant mets by the KellyCis83 cells and their lymph node colonisation may be partially explained by the increased CCL2 expression. This is an important research

question because a growing body of evidence suggests that both tumour and immune cells and their soluble agents can influence the process of metastatic dissemination. In the context of neuroblastoma, further investigations are needed to decode the CCL2-CCR2 axes, and our model may be a valuable tool.

Our study has identified the tumour necrosis factor receptor superfamily member 4 (*TNFRSF4*) gene as a marker of interest in neuroblastoma, particularly when looking at high-risk disease. *TNFRSF4* was found to be significantly downregulated in metastasis established by the cisplatin-resistant cell line KellyCis83Luc compared to its cisplatin-sensitive counterpart KellyLuc. We demonstrated that Low expression of *TNFRSF4* was significantly associated with reduced probability of both event-free and overall survival in neuroblastoma and as such TNFRSF4 can be considered as an independent prognostic factor. Although the expression of this gene shows inverse correlations with the expression of the *MYCN* oncogene, our studies using the SHEP-Tet21N inducible system concluded that *TNFRSF4* is not directly regulated by MYCN.

The prognostic power of *TNFRSF4* has already been studied in several cancers, with some contrasting results. Overexpression has been found to be unfavourable in head and neck squamous cell carcinoma [36], acute myeloid leukaemia (AML) [37] and hepatocellular carcinoma (HCC) [38]. While these prognostic trends conflict with the trends observed in our model, several studies have also found that elevated expression of *TNFRSF4* and its protein counterpart OX40 as favourable in cancers, including colorectal cancer (CRC) [39,40], cutaneous malignant melanoma [41] and endometrial carcinoma (EC) [42]. Many of these studies have listed *TNFRSF4* as a promising potential prognostic indicator, either alone or as part of a gene panel. It would therefore be compelling to continue to clinically study the association between this marker and survival rates in neuroblastoma and identify related genes that may aid in the establishment of a prognostic gene panel or combine it with well-established prognostic genes such as *ALK*, *PHOX2B*, and *PTPRD* [43].

In our model, we observed widespread migration of neuroblastoma cells to the lymph nodes, with significantly more metastases developing in the cisplatin-resistant group. In other cancers, including melanoma, carcinomas and breast cancer, the expression of *TNFRSF4* in the lymph nodes has been previously assessed. Miragaia et al. (2019) found that TNFRSF-NF-kB-related genes, including *TNFRSF1B* and *TNFRSF4*, were upregulated in the lymph nodes, playing a key role in effector Treg cell development [44]. Vetto et al. found that OX40 was expressed by 28% of the draining lymph node cells in carcinoma patients, and expression was higher in tumour infiltrating T cells than peripheral blood T cells [45]. Ramstad et al. found that in breast cancer, OX40 expression was elevated in 45% of samples resected from axillary lymph node metastases due to upregulation of lymphocytes within tumour-draining lymph nodes [46]. This is supported by findings from Xie et al. (2010), who demonstrated a significant association between elevated OX40 expression and lymph node metastasis in breast cancer [47]. Similarly, in metastatic melanoma, sentinel lymph nodes had a significantly higher number of OX40+ lymphocytes than healthy lymph nodes. These findings appear to conflict with our observation of decreased *TNFRSF4* expression in cisplatin-resistant lymph node metastases in neuroblastoma; however, many of these studies assessed *TNFRSF4* expression in lymphocytes within tumour-positive lymph nodes rather than in tumour cells themselves, as we did in our study. Further studies in immune-competent mice will allow to discern whether the inverse correlation between the immune cells and tumour cells is necessary and/or sufficient to permit the metastatic growth.

Adding to *TNFRSF4*'s potential role as a prognostic indicator in cancer, the interaction between its produced protein OX40 and ligand OX40L has been proposed as a potential target for immunotherapy, as the resultant signals can promote survival of CD4+ and CD8+ T cells,

sustain their anti-apoptotic protein expression and enhance cytokine production to augment tumour-specific T-cell response [48]. In neuroblastoma, Thakur et al found that OX40 expression was strikingly low in tumour tissue from a group of 100 paediatric cancers, including neuroblastoma, Ewing's sarcoma, medulloblastoma, osteosarcoma and rhabdomyosarcoma [49]. Long et al studied the incorporation of OX40 and CD28 costimulatory domains in GD2-specific CAR T cells and found that they mediated efficient and comparable lysis of both GD2+ sarcoma and neuroblastoma cell lines *in vitro* [50]. Similarly, Quintarelli et al 2018 examined [46] these third-generation CAR T cells and found that CARs incorporating a CD28 costimulatory domain are associated with superior antitumour efficacy in clinical trials compared to those incorporating the combination of CD28 and OX40 domains [51]. The most recent single cell RNA sequencing of neuroblastoma pre- and post- chemotherapy samples identified OX40 in tumour-infiltrating Tregs suggesting their enhanced suppressive capacity [52].

OX40 agonists have already demonstrated success and are well tolerated in preclinical studies and in early phase human trials in metastatic tumours [53,54]. The success of these agonists is dependent on tumour immunogenicity for sufficient priming signals, which raises concerns about their potential in poorly immunogenic neuroblastoma. However, it is possible to sensitise poorly immunogenic tumours prior to administration of OX40 agonists by intratumoural CD4+ lymphodepletion, as has been shown in melanoma. This provides hope that this strategy could be implemented in other poorly immunogenic tumours, particularly when it is considered that neuroblastoma and melanoma share the same progenitor cells derived from the neural crest.

## Conclusion

This study described an *in vivo* cisplatin-resistant metastatic model of neuroblastoma. This model represents a metastatic disease that colonises the lymph nodes–one of the most common sites of neuroblastoma metastasis (31%) [4]. It is supplemented with the immune-oncology transcriptomic dataset that can guide further research into the role of the lymphatic system in neuroblastoma metastasis formation. While murine models cannot replace clinical trials, there is a growing demand for informative tools that can uncover the molecular signalling mechanisms. Since our model exhibits a drug-resistant and predictive metastatic phenotype, it can serve as a valuable preclinical tool for evaluating the efficacy of (new) therapeutic agents and their combinations for neuroblastoma patients with metastatic disease.

This preclinical model led to the identification of significant dysregulation of TNFRSF4 in cisplatin-resistant tumours. We demonstrated the significance of this gene in patient survival studies in a large cohort of neuroblastomas and showed a connection between the expression of this marker and other well-established prognostic indicators of neuroblastoma. The exact mechanisms regulating *TNFRSF4* expression remain uncertain.

## Supporting information

**S1 Fig. Mouse weight variation in the murine xenograft model.**
(PDF)

**S2 Fig. Cisplatin-resistant KellyCis83Luc tail-vein-injected xenografts have more metastatic foci than drug-sensitive KellyLuc xenografts.**
(PDF)

**S3 Fig. Haematoxylin and eosin staining of tumours resected from the drug-resistant neuroblastoma xenograft model.**
(PDF)

**S4 Fig. Metascape pathway and process enrichment analysis.**
(PDF)

**S5 Fig. Kaplan–Meier survival curves for shortlisted genes that demonstrated clinical significance.**
(PDF)

**S6 Fig. RT-qPCR validation of TNFRSF4 expression trends in KellyLuc and KellyCis83Luc cells.**
(PDF)

**S7 Fig.** Correlation between MYCN and TNFRSF4 expression generated in R2GAVP (22) using the Westermann cohort of 579 neuroblastomas (A) and the Fischer cohort of 223 neuroblastoma (B).
(PDF)

**S8 Fig. Expression of TNFRSF4 and MYCN in a panel of neuroblastoma cell lines.**
(PDF)

**S9 Fig. Expression of CCL2 in a cisplatin-resistant metastatic model of neuroblastoma.**
(PDF)

**S1 Table. Available identifiers for R2 tumour neuroblastoma cohorts.**
(PDF)

## Acknowledgments

We thank John Nolan, PhD for his technical support of the sample preparation for the HTG EdgeSeq.

## Author Contributions

**Conceptualization:** Miguel F. Segura, Olga Piskareva.

**Data curation:** Laura Devis-Jauregui, Aroa Soriano Fernandez, Josep Roma, Miguel F. Segura, Olga Piskareva.

**Formal analysis:** Catherine Murphy, Laura Devis-Jauregui, Ronja Struck, Ariadna Boloix, Ciara Gallagher, Cian Gavin, Stephen Madden, Josep Roma, Miguel F. Segura, Olga Piskareva.

**Funding acquisition:** Laura Devis-Jauregui, Ronja Struck, Ciara Gallagher, Miguel F. Segura, Olga Piskareva.

**Investigation:** Catherine Murphy, Laura Devis-Jauregui, Ronja Struck, Ciara Gallagher, Cian Gavin, Federica Cottone, Aroa Soriano Fernandez, Miguel F. Segura, Olga Piskareva.

**Methodology:** Catherine Murphy, Laura Devis-Jauregui, Ariadna Boloix, Cian Gavin, Federica Cottone, Aroa Soriano Fernandez, Miguel F. Segura, Olga Piskareva.

**Project administration:** Olga Piskareva.

**Resources:** Josep Roma, Miguel F. Segura, Olga Piskareva.

**Software:** Laura Devis-Jauregui, Stephen Madden, Olga Piskareva.

**Supervision:** Miguel F. Segura, Olga Piskareva.

**Validation:** Catherine Murphy, Federica Cottone, Aroa Soriano Fernandez, Miguel F. Segura, Olga Piskareva.

**Visualization:** Catherine Murphy, Ronja Struck, Aroa Soriano Fernandez, Miguel F. Segura, Olga Piskareva.

**Writing – original draft:** Catherine Murphy, Miguel F. Segura, Olga Piskareva.

**Writing – review & editing:** Catherine Murphy, Laura Devis-Jauregui, Ronja Struck, Ariadna Boloix, Aroa Soriano Fernandez, Miguel F. Segura, Olga Piskareva.

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
