## [Decision Letter · Decision Letter 0]

26 Feb 2024

PONE-D-24-02305In vivo cisplatin-resistant neuroblastoma metastatic model reveals Tumour Necrosis Factor Receptor Superfamily Member 4 (TNFRSF4) as an independent prognostic factor of survival in neuroblastomaPLOS ONE

Dear Dr. Piskareva,

Thank you for submitting your manuscript to PLOS ONE. After careful consideration, we feel that it has merit but does not fully meet PLOS ONE’s publication criteria as it currently stands. Therefore, we invite you to submit a revised version of the manuscript that addresses the points raised during the review process.

We look forward to receiving your revised manuscript.

Kind regards,

Seok-Geun Lee, PhD

Academic Editor

PLOS ONE

Journal Requirements:

"O.P. received support for this project through Neuroblastoma UK, RCSI Strategic Academic Recruitment (StAR) Programme, Health Research Board - The Conor Foley Neuroblastoma Cancer Research Foundation. C.G. was funded by Irish Research Council Postgraduate Programme (GOIPG/2019/3220), R.S. - Irish Research Council - The Conor Foley Neuroblastoma Cancer Research Foundation (EPSPG/2021/95). M.S. was funded by Instituto de Salud Carlos III through the projects “PI20/000530”, “PI23/01144” and “PMP21/00073” (Co-funded by the European Regional Development Fund/European Social Fund; “A way to make Europe”/ “Investing in your future”). L.D.-J. is recipient of a Ramón y Cajal scheme (Grant No. RyC-2021-034346-I), funded by the Spanish Ministry for Science and Innovation (MCIN)"

Please state what role the funders took in the study.  If the funders had no role, please state: ""The funders had no role in study design, data collection and analysis, decision to publish, or preparation of the manuscript."" If this statement is not correct you must amend it as needed. 

5. Please be informed that funding information should not appear in the Acknowledgments section or other areas of your manuscript. We will only publish funding information present in the Funding Statement section of the online submission form. Please remove any funding-related text from the manuscript. 

7. Your ethics statement should only appear in the Methods section of your manuscript. If your ethics statement is written in any section besides the Methods, please delete it from any other section. 

**Additional Editor Comments:**

Both reviewers agreed the manuscript to be interesting. However, they also raised concerns regarding the presentation, analysis, and interpretation of the data. They also noted a lack of data and discussion to fully support the authors' hypothesis and claims made in the manuscript. Additionally, the reviewers identified issues with some of the experimental designs, including the animal model. It is important to address each of these concerns meticulously and provide successful responses.

Reviewers' comments:

Reviewer's Responses to Questions

**Comments to the Author**

1. Is the manuscript technically sound, and do the data support the conclusions?

Reviewer #1: Yes

Reviewer #2: Partly

2. Has the statistical analysis been performed appropriately and rigorously? 

Reviewer #1: Yes

Reviewer #2: No

3. Have the authors made all data underlying the findings in their manuscript fully available?

Reviewer #1: Yes

Reviewer #2: No

4. Is the manuscript presented in an intelligible fashion and written in standard English?

Reviewer #1: Yes

Reviewer #2: Yes

5. Review Comments to the Author

Reviewer #1: This manuscript utilizes a murine model of neuroblastoma to compare a parental cell line versus its cisplatin resistant derivative to assess a possible impact on metastasis. The authors discover a disproportional metastatic tumor load associated with cisplatin resistance and identify dysregulated genes in these tumors associated with clinical relevance. The gene TNFRSF4 was shown to correlate with poor prognostic outcomes, though not risk of recurrence. Data indicated that TNFRSF4 did not appear to be regulated by changes in MYCN expression and the knockdown of TNFRSF4 expression did not significantly affect cell viability in vitro.

Here are my comments:

Major Comments:

1) On page 19, the authors indicate the following in regard to multivariate Cox regression analysis:

“The increased likelihood of recurrence based on Low expression of TNFRSF4 was no longer significant in this analysis, but this variable remained significant at increasing the likelihood of death 1.7-fold (p=0.007). Then, TNFRSF4 Low expression was an independent prognostic factor of survival in neuroblastoma.”

If TNFRSF4 is no longer considered significant to the likelihood of recurrence, what role would you speculate TNFRSF4 is responsible for in the increased likelihood of death?

One would assume this role must be different from the roles of the remaining hazards, which all correlated with both “Risk of Recurrence” and “Risk of Death”.

2) In Figure 7 (Section 3.7, page 22), the authors indicated that siRNA knockdown of KIF11 in both Kelly and SK-N-AS neuroblastoma cells served as a positive control for cell viability and proliferation. Figure 7D corroborated this result. However, siRNA knockdown of TNFRSF4 (confirmed in Figure 7E) revealed minimal effects in the neuroblastomas. Unfortunately, that leaves the manuscript with no functional assay and therefore, no data to suggest what role TNFRSF4 plays in the predicted “increased likelihood of death” witnessed in patients. Can the authors offer any assay which helps point in the direction of a possible mechanism of action?

3) As a follow-up to Comment #2, given the volume of RNA Sequencing data you have acquired and the use of pathway analysis performed (using iDEP, for instance), could this data help to identify a relevant dysregulated pathway for a predicted function of TNFRSF4 in neuroblastoma mortality?

4) In the first paragraph of the Discussion (page 24), the authors indicate:

“Our study identified a potential relationship between the drug-resistant phenotype, which causes relapse in approximately 50% of neuroblastomas, the formation of a sentinel lymph node microenvironment during tumour metastasis and immune-related gene expression alterations”.

I apologize, but can you clarify exactly what “immune-related gene expression alterations” were covered in this manuscript so far?

Section 3.4 (page 15 – 16) indicates the title:

“Immune-Related Genes are Associated with Neuroblastoma Outcomes”

This section mentions “candidate genes” associated with “neuroblastoma aggressiveness” and assumes “a direct neuroblastoma – immune cell interaction would be facilitated by either proteins secreted or expressed on the cancer cell surface”. However, the reader has no indication exactly how these candidates are immune related. Since this section deals with “Immune-Related Genes”, shouldn’t we know how they are immune related?

5) The Discussion further mentions (bottom of page 24):

“Pathway enrichment analysis of the profiled immuno-oncology panel via iDEP and Metascape points towards some dysregulated genes that can modulate the formation of metastatic sites through cytokine signalling”.

The text then identifies the cytokine CCL2 as upregulated in cisplatin-resistant tumors and refers to a supplemental figure. Why was this new data only introduced in the Discussion? This is an experimental result. Shouldn’t it have been mentioned in Section 3.4?

6) In the Discussion (page 26), the authors identify that TNFRSF4 “upregulated in the lymph nodes, playing a key role in effector Treg cell development”, that “expression was higher in tumour infiltrating T cells than peripheral blood T cells“, and “in metastatic melanoma, sentinel lymph nodes had a significantly higher number of OX40+ lymphocytes than healthy lymph nodes expression was higher in tumour infiltrating T cells than peripheral blood T cells“.

The authors further indicate that these data “conflict with our observation of decreased TNFRSF4 expression in cisplatin-resistant lymph node metastases in neuroblastoma; however, many of these studies assessed TNFRSF4 expression in lymphocytes within tumour-positive lymph nodes rather than in tumour cells themselves, as we did in our study.“

Given this inversed proportion of expression for TNFRSF4 between lymph node lymphocytes and the tumors themselves, is this correlation necessary or advantageous and why would you propose this is occurring?

Minor Comments:

1) I am curious why the authors chose SCID Beige mice as a model system for this study. Were nude athymic mice not viable for metastatic modeling?

2) On page 4 (Section 2.2 of the Materials and Methods), the authors indicate the following:

“Fox Chase SCID Beige mice, which have defective B cells, T cells and NK cells, were injected via tail vein with a suspension of either KellyLuc (cisplatin-sensitive, n=10) or KellyCis83Luc (cisplatin-resistant, n=10) cells.”

Could you indicate how many cells were introduced, the volume, and the characteristics of the “suspension” solution?

3) Figure 3 (associated with Section 3.2, page 15) appears to have been incorrectly labeled as Figure 1. It is also incorrectly labeled in the text as Figure 1A on page 14 and Figures 1B, C, D, and E on page 16. The actual Figure 1 is identified in Section 3.1 (page 12).

4) On page 21 (Section 3.6), the authors state:

“Owing to the significant inverse correlation shown between MYCN and TNFRSF4 in multiple neuroblastoma datasets (Figure 7A and Supplementary Figure S7)…”

I believe you are referring to Supplementary Figure S6, titled “Correlation between MYCN and TNFRSF4 expression…” which indicates the comparison of the Westermann cohort versus the the Fischer cohort of neuroblastomas.

Supplementary Figure S7 is titled: “Expression of TNFRSF4 and MYCN in a panel of neuroblastoma cell lines.”

As a consequence, Supplementary Figure 6 is not mentioned anywhere in the text and Supplementary Figure S7 needs to be properly introduced.

5) On page 25 (in the Discussion), Supplemental Figure S9 is indicated as associated with an experiment involving CCL2. There is no Supplemental Figure S9 in the supplementary material. This is Supplemental Figure S8.

6) In the Discussion (page 25), the following statement should probably have a citation to support it:

“The CCL2-CCR2 pair has a long track record of multiple protumorigenic roles, including the promotion of tumour growth and metastasis in many cancer types.”

Taken together, this manuscript offers a solid set of experiments, but requires a few gaps and corrections be addressed.

Reviewer #2: This manuscript focuses on further characterization of an established cisplastin-resistant neuroblastoma model (KellyCis83), in the context of metastasis, an important consequence and deadly outcome for many patients with high-risk NB. Given that metastatic NB is highly drug resistant, the in vivo RNA sequencing dataset and assessments made regarding TNFRSF4 would provide a valuable resource to the research community.

One concern is the conclusion that TNFRSF4 disregulation is specific to NB tumor cells. Bulk tumor was dissected, stained and profiled using RNA sequencing. Why is it assumed that leukocytes and/or other stromal cells are not perhaps an additional and/or functional the source of TNFRSF4 in the in vivo model? Please clarify with tissue specific expression profiling from the met biopsies.

Related to the previous comment regarding the potential for TNFRSF4 expression in stromal tissues, scRNAseq data from NB patients (Wienke et al. Cancer Cell 2024) show TNFRSF4 expression in NB-associated Tregs and potentially an immune suppressive activity by this cell type. This recently published data might lead one to guess that increased stromal TNFRSF4 expression associates with worsened outcomes, which contradicts the data shown in Fig 4. It may be a consideration to discuss this published finding as it relates to the current manuscript, KellyCis83 and TNFRSF4.

Even though characterization of these Kelly cell lines was previously published, it would be beneficial to comment in section 3.1 or elsewhere on differences between Kelly and KellyCis83Luc with respect to growth dynamics and genomic changes (chromosomal gains, losses, rearrangements). For example, are any changes consistent with chromosomal loss discussed in patients on pg. 18?

Is there an explanation for why the RT-PCR data (pg. 15, Fig. 1 C and E) are inconsistent with the RNA sequencing. Is it that these genes differentially expressed in stromal cells? This could be further explored on tumor section and/or clarified in the manuscript.

In the discussion, it is mentioned that the tumor size of KellyCis83 mets are increased. It would be interesting to present this data in Fig 1 to compliment met number.

It does not look like the stiffness data presented in Fig.1F is significantly different in Kelly versus KellyCis83?. While this may be a possible explanation for the observed phenotypic changes, without significant changes, this reviewer does not fully understand the value of presenting and discussing this data in the manuscript. A p-value in Fig.1F may help clarify this point.

Minor comments:

-"Figure 1: Functional enrichment analysis..." (pg.15) figure legend and associated references in the text appear mislabelled and based on the presented sequence should be Figure 3 vs Figure 1?

-A legend for "dot sizes correspond to adjusted p values" in Fig1 (maybe Fig.3, pg 15) could be helpful

-Reference the source of "large cohort of 498 neuroblastomas" on page 16, line 2.

-Raw data from the RNA sequencing do not appear to be deposited in a publically available database. Sharing of these data would facilitate their use in other studies.

6. PLOS authors have the option to publish the peer review history of their article (what does this mean?). If published, this will include your full peer review and any attached files.

Reviewer #1: No

Reviewer #2: No

---

## [Author Response · Author response to Decision Letter 0]

10 Apr 2024

Dear Editor and Reviewers,

On behalf of all co-author, I would like to thank the reviewers for their time and expertise in evaluating our manuscript. I appreciate the comprehensive and constructive appraisal of our work and positive comments.

I hope that Reviewer’s concerns have been addressed and added more clarity to the submitted manuscript.

All our responses are in red below. A corresponding reference to the manuscript page is provided. All changes in the manuscript body are tracked. A pdf version of the same is also attached.

Yours sincerely,

Olga Piskareva

Corresponding author.

Journal Requirements:

We have checked this information and revised the manuscript accordingly.

We have checked this information.

We have checked this information and revised the manuscript accordingly.

"O.P. received support for this project through Neuroblastoma UK, RCSI Strategic Academic Recruitment (StAR) Programme, Health Research Board - The Conor Foley Neuroblastoma Cancer Research Foundation (HRCI-HRB-2022-013). C.G. was funded by Irish Research Council Postgraduate Programme (GOIPG/2019/3220), R.S. - Irish Research Council - The Conor Foley Neuroblastoma Cancer Research Foundation (EPSPG/2021/95). M.S. was funded by Instituto de Salud Carlos III through the projects “ICI21/00076”, “PI23/01144” and “PMP21/00073” (Co-funded by the European Regional Development Fund/European Social Fund; “A way to make Europe”/ “Investing in your future”). L.D.-J. is recipient of a Ramón y Cajal scheme (Grant No. RyC-2021-034346-I), funded by the Spanish Ministry for Science and Innovation (MCIN)"

Please state what role the funders took in the study. If the funders had no role, please state: ""The funders had no role in study design, data collection and analysis, decision to publish, or preparation of the manuscript."" If this statement is not correct you must amend it as needed. We have added this statement and updated the grant numbers, p28.

5. Please be informed that funding information should not appear in the Acknowledgments section or other areas of your manuscript. We will only publish funding information present in the Funding Statement section of the online submission form. Please remove any funding-related text from the manuscript. 

We have checked the manuscript and confirmed that funding information appears in the Funding Statement section.

We made the HTG EdgeSeq immuno-oncology panel mRNA profiling data available at R2: Genomics Analysis and Visualization Platform. P 28. 

7. Your ethics statement should only appear in the Methods section of your manuscript. If your ethics statement is written in any section besides the Methods, please delete it from any other section. 

We removed the duplication from p 28. The ethics statement appears in the Methods section.

We have checked this information and revised the manuscript accordingly.

Additional Editor Comments:

Both reviewers agreed the manuscript to be interesting. However, they also raised concerns regarding the presentation, analysis, and interpretation of the data. They also noted a lack of data and discussion to fully support the authors' hypothesis and claims made in the manuscript. Additionally, the reviewers identified issues with some of the experimental designs, including the animal model. It is important to address each of these concerns meticulously and provide successful responses.

Reviewers' comments:

Reviewer's Responses to Questions

Comments to the Author

1. Is the manuscript technically sound, and do the data support the conclusions?

Reviewer #1: Yes

Reviewer #2: Partly

2. Has the statistical analysis been performed appropriately and rigorously?

Reviewer #1: Yes

Reviewer #2: No

3. Have the authors made all data underlying the findings in their manuscript fully available?

Reviewer #1: Yes

Reviewer #2: No

4. Is the manuscript presented in an intelligible fashion and written in standard English?

Reviewer #1: Yes

Reviewer #2: Yes

5. Review Comments to the Author

Reviewer #1: This manuscript utilizes a murine model of neuroblastoma to compare a parental cell line versus its cisplatin resistant derivative to assess a possible impact on metastasis. The authors discover a disproportional metastatic tumor load associated with cisplatin resistance and identify dysregulated genes in these tumors associated with clinical relevance. The gene TNFRSF4 was shown to correlate with poor prognostic outcomes, though not risk of recurrence. Data indicated that TNFRSF4 did not appear to be regulated by changes in MYCN expression and the knockdown of TNFRSF4 expression did not significantly affect cell viability in vitro.

Here are my comments:

Major Comments:

1) On page 19, the authors indicate the following in regard to multivariate Cox regression analysis:

“The increased likelihood of recurrence based on Low expression of TNFRSF4 was no longer significant in this analysis, but this variable remained significant at increasing the likelihood of death 1.7-fold (p=0.007). Then, TNFRSF4 Low expression was an independent prognostic factor of survival in neuroblastoma.”

If TNFRSF4 is no longer considered significant to the likelihood of recurrence, what role would you speculate TNFRSF4 is responsible for in the increased likelihood of death?

One would assume this role must be different from the roles of the remaining hazards, which all correlated with both “Risk of Recurrence” and “Risk of Death”. 

We have carefully considered the reviewer comment and provide our reasoning below.

Kaplan-Meier analysis evidenced that Low expression of TNFRSF4 significantly reduced survival probabilities in neuroblastoma, in the SEQC cohort of 498 neuroblastomas, in terms of both OS (p=1.53e-09) and EFS (p=4.2e-04) (Figure 4A-B of the manuscript). 

As stated by the reviewer, once the relation between Low TNFRSF4 expression and patient outcome was evidenced, additionally we performed Cox univariate and multivariate analysis to determine if TNFRSF4 is an independent predictor of EFS and OS within the studied population.

In the univariate analysis, TNFRSF4 HR was statistically significant for both recurrence and death (Figure 5 of the manuscript). 

However, the marker is a more powerful predictor of the death event (OS), as can be observed from the ROC curves for both OS and EFS:

For the OS, the is AUC= 0.657 (p= 7.8 x10-7):

For the EFS, AUC= 0.564 (p= 0.018):

This is also why, the Low/High TNFRSF4 expression for the analysis was generated based on Youden Index for the OS event.

Cox regression analysis allows for a better understanding of the predictive value of the marker by comparing it to the basic clinical variables defined for the patient risk: MYCN amp/no-amp, age, and ISSN stage. As a survival marker (OS), it provides a specific term to the multivariate Cox constructed model, thus improving the survival prognosis made from the above mentioned three basic clinical variables. However, as a marker of event (EFS), its value is more limited since this global potential observed on a univariate basis is not maintained at a global level (p=0.211) when studied in a multivariate context that includes MYCN amp/no-amp.

In fact, the three bivariate Cox regression analysis with TNFRSF4 and the three clinical variables already mentioned have different behaviors: when including MYCN amp/no-amp, the model does not retain the TNFRSF4 variable although it is close to significance (p=0.083). TNFRSF4 is retained in the regression analysis model when including both the age (p<0.001) or the ISSN stage (p<0.001).

Tables illustrating these results can be shown as follows:

Bivariate regression analysis (TNFRSF4 & Age) 

Factors Event free survival (recurrence)

 HR (95% CI) P value

TNFRSF4 (Low vs High) 1.439 (1.067-1.940) 0.017

Age (≥18 months vs <18 months) 2.899 (3.937-2.141) 0.000

Abbreviations: HR, hazard ratio 

Bivariate regression analysis (TNFRSF4 & ISSN stage) 

Factors Event free survival (recurrence)

 HR (95% CI) P value

TNFRSF4 (Low vs High) 1.423 (1.056-1.918) 0.021

ISSN Stage (4 vs others) 3.726 (2.746-5.056) 0.000

Abbreviations: HR, hazard ratio 

Bivariate regression analysis (TNFRSF4 & MYCN amp no-amp) 

Factors Event free survival (recurrence)

 HR (95% CI) P value

TNFRSF4 (Low vs High) 1.319 (0.964-1.806) 0.084

MYCN (amp vs no-amp) 2.941 (2.113-4.095) 0.000

Abbreviations: HR, hazard ratio 

In this cohort, regarding the Cox multivariate regression model, MYCN amp/no-amp variable would be the cause why TNFRSF4 is not retained by the model. The two variables show interrelation in the study. However, more studies in larger cohorts, would be required to deep in the role of the gene of interest as a marker of tumour event in neuroblastoma.

2) In Figure 7 (Section 3.7, page 22), the authors indicated that siRNA knockdown of KIF11 in both Kelly and SK-N-AS neuroblastoma cells served as a positive control for cell viability and proliferation. Figure 7D corroborated this result. However, siRNA knockdown of TNFRSF4 (confirmed in Figure 7E) revealed minimal effects in the neuroblastomas. Unfortunately, that leaves the manuscript with no functional assay and therefore, no data to suggest what role TNFRSF4 plays in the predicted “increased likelihood of death” witnessed in patients. Can the authors offer any assay which helps point in the direction of a possible mechanism of action? 

While the standard viability and proliferation assays were insufficient to conclude on the biological function of OX40 in neuroblastoma cells, they did provide a reference point to search for other potential roles. Other test that could link TNFRSF4 with aggressiveness could be changes in the interaction with immune cells. For example:

• Changes in the sensitivity to death ligands (FasL, TNFalpha)

• Changes in the activity of the NFkB pathway, that usually discriminates between pro-survival and pro-death signals.

• Changes in the released cytokine profile.

While we believe that this could be an interesting experiment it is rather out of scope of the present work. We apologise for not giving more insights in this matter and we added a sentence in the discussion to encourage future work to answer this question. P22

3) As a follow-up to Comment #2, given the volume of RNA Sequencing data you have acquired and the use of pathway analysis performed (using iDEP, for instance), could this data help to identify a relevant dysregulated pathway for a predicted function of TNFRSF4 in neuroblastoma mortality?

Unfortunately, the volume of acquired data is limited. We used a predefined panel of immune-oncology markers (549 genes and only 36 genes were significantly dysregulated). 36 genes is not sufficient to predict a relevant dysregulated pathway for a predicted function of TNFRSF4 in neuroblastoma mortality.

4) In the first paragraph of the Discussion (page 24), the authors indicate:

“Our study identified a potential relationship between the drug-resistant phenotype, which causes relapse in approximately 50% of neuroblastomas, the formation of a sentinel lymph node microenvironment during tumour metastasis and immune-related gene expression alterations”.

I apologize, but can you clarify exactly what “immune-related gene expression alterations” were covered in this manuscript so far?

Thank you for this observation, which may be confusing. We revised this phrase throughout the text and replaced with assessed immune-oncology genes

Section 3.4 (page 15 – 16) indicates the title:

“Immune-Related Genes are Associated with Neuroblastoma Outcomes”

This section mentions “candidate genes” associated with “neuroblastoma aggressiveness” and assumes “a direct neuroblastoma – immune cell interaction would be facilitated by either proteins secreted or expressed on the cancer cell surface”. However, the reader has no indication exactly how these candidates are immune related. Since this section deals with “Immune-Related Genes”, shouldn’t we know how they are immune related?

We agree that this section needs more wording clarity. We revised this section and referred to our immune-oncology gene panel

5) The Discussion further mentions (bottom of page 24):

“Pathway enrichment analysis of the profiled immuno-oncology panel via iDEP

---

## [Decision Letter · Decision Letter 1]

30 Apr 2024

In vivo cisplatin-resistant neuroblastoma metastatic model reveals Tumour Necrosis Factor Receptor Superfamily Member 4 (TNFRSF4) as an independent prognostic factor of survival in neuroblastoma

PONE-D-24-02305R1

Dear Dr. Piskareva,

We’re pleased to inform you that your manuscript has been judged scientifically suitable for publication and will be formally accepted for publication once it meets all outstanding technical requirements.

Kind regards,

Seok-Geun Lee, PhD

Academic Editor

PLOS ONE

Additional Editor Comments (optional):

Reviewers' comments:

Reviewer's Responses to Questions

**Comments to the Author**

1. If the authors have adequately addressed your comments raised in a previous round of review and you feel that this manuscript is now acceptable for publication, you may indicate that here to bypass the “Comments to the Author” section, enter your conflict of interest statement in the “Confidential to Editor” section, and submit your "Accept" recommendation.

Reviewer #1: All comments have been addressed

Reviewer #2: All comments have been addressed

2. Is the manuscript technically sound, and do the data support the conclusions?

Reviewer #1: Yes

Reviewer #2: Yes

3. Has the statistical analysis been performed appropriately and rigorously? 

Reviewer #1: Yes

Reviewer #2: Yes

4. Have the authors made all data underlying the findings in their manuscript fully available?

Reviewer #1: Yes

Reviewer #2: Yes

5. Is the manuscript presented in an intelligible fashion and written in standard English?

Reviewer #1: Yes

Reviewer #2: Yes

6. Review Comments to the Author

Reviewer #1: This manuscript identified a gene which correlates with poor outcome for high-risk neuroblastoma patients based upon a murine metastasis study. The material presented is sound and the analysis does not exaggerate its impact or conclusions. Therefore, I recommend its acceptance for publication.

Reviewer #2: Thank you to the authors for their thoughtful response and updated manuscript.

All of my questions and comments have been addressed.

7. PLOS authors have the option to publish the peer review history of their article (what does this mean?). If published, this will include your full peer review and any attached files.

Reviewer #1: No

Reviewer #2: No

---

## [Editor Report · Acceptance letter]

17 May 2024

PONE-D-24-02305R1 

PLOS ONE

Dear Dr. Piskareva, 

I'm pleased to inform you that your manuscript has been deemed suitable for publication in PLOS ONE. Congratulations! Your manuscript is now being handed over to our production team.

Kind regards, 

on behalf of

Dr. Seok-Geun Lee 

Academic Editor

PLOS ONE